# Species abundance correlations carry limited information about microbial network interactions

**Susanne Pinto**[1]*, **Elisa Benincà**[2], **Egbert H. van Nes**[3], **Marten Scheffer**[3], **Johannes A. Bogaards**[4]

**1** Department of Biomedical Data Sciences, Leiden University Medical Center, Leiden, The Netherlands, **2** Centre for Infectious Disease Control, National Institute for Public Health and the Environment (RIVM), Bilthoven, The Netherlands, **3** Department of Environmental Sciences, Wageningen University and Research, Wageningen, The Netherlands, **4** Department of Epidemiology and Data Science, Amsterdam UMC location Vrije Universiteit Amsterdam, Amsterdam, Netherlands

* s.pinto@lumc.nl

**Data Availability Statement:** All relevant codes are available via GitHub (https://github.com/susannepinto/gLV_microbiome.git).

## Abstract

Unraveling the network of interactions in ecological communities is a daunting task. Common methods to infer interspecific interactions from cross-sectional data are based on co-occurrence measures. For instance, interactions in the human microbiome are often inferred from correlations between the abundances of bacterial phylogenetic groups across subjects. We tested whether such correlation-based methods are indeed reliable for inferring interaction networks. For this purpose, we simulated bacterial communities by means of the generalized Lotka-Volterra model, with variation in model parameters representing variability among hosts. Our results show that correlations can be indicative for presence of bacterial interactions, but only when measurement noise is low relative to the variation in interaction strengths between hosts. Indication of interaction was affected by type of interaction network, process noise and sampling under non-equilibrium conditions. The sign of a correlation mostly coincided with the nature of the strongest pairwise interaction, but this is not necessarily the case. For instance, under rare conditions of identical interaction strength, we found that competitive and exploitative interactions can result in positive as well as negative correlations. Thus, cross-sectional abundance data carry limited information on specific interaction types. Correlations in abundance may hint at interactions but require independent validation.

## Author summary

The bacteria in and on our body (the human microbiome) largely determine how our body functions, and whether we stay healthy or get sick. These bacteria do not live on their own, but interact among each other and with their human host. Finding out which bacteria interact with each other is cumbersome, but patterns of joint occurrence between species might provide a clue to their ecological dependencies. We investigated whether correlations in species abundance can be used for the purpose of ecological network reconstruction. We simulated different bacterial communities with known interactions

**Funding:** This publication is part of the project "Ecology meets human health: unraveling the complex dynamics of human microbiota to direct therapeutic intervention" financed by the Dutch Organization for Scientific Research (NWO) through the research program Complexity in Health and Nutrition (NWO grant 645.001.002; www.nwo.nl/onderzoeksprogrammas/complexiteit), with co-funding by the National Institute for Public Health and the Environment (RIVM) of the Netherlands. The funders had no role in study design, data collection and analysis, decision to publish, or preparation of the manuscript.

**Competing interests:** The authors have declared that no competing interests exist.

according to a theoretical population model. After having collected virtual samples from our simulated data, we performed a correlation analysis and then compared the correlation network with our known interaction network. We found that correlations can be informative for underlying interactions, but ecological conclusions should be drawn carefully. An obvious limitation of correlation analysis is that direction of interaction cannot be recovered from co-occurrence data, making correlations insensitive for detection of asymmetric interactions. In addition, we found that competitive and exploitative interactions can induce positive as well as negative correlations. We recommend careful interpretation and validation when inferring networks from cross-sectional abundance data.

## Introduction

The human body harbors an exceptional bacterial diversity [1]. The composition of these bacterial communities is generally shaped by characteristics of the host and by the ecological dependencies among bacterial species themselves [2–4]. These dependencies often occur through competitive or synergistic interactions, which may lead to a (mutual) decrease or increase in the abundance of interacting species [5]. For instance, it is known that bacteria can interact with each other through excreted metabolites, which can function as an antimicrobial or as a food source [2,6]. Among other mechanisms, for example, negative interactions take place when toxic compounds produced by one species harm other bacteria, whereas positive interactions occur when bacteria feed on the nutrients that are produced by others. Besides, many different forms of interactions exist, depending on the effects experienced by the species involved. Knowledge of interspecific interactions in the human microbiome is paramount to understand ecological processes and compositional changes in relation to health and disease [7,8].

Most human microbiome studies are limited to only a few samples in time, presenting mere 'snapshots' of the microbial ecosystem, even if these are derived from hundreds of human hosts. A common way to infer microbial networks from such cross-sectional data is by quantifying co-occurrence, e.g., through (partial) correlations, between bacterial phylogenetic groups. Several different conclusions have been derived from such endeavors, for example on species associations that reflect shared or overlapping niche preferences [9], microbial community structure [10,11], the resilience of microbial communities to perturbations [12] and keystone species in microbial networks [13]. Currently there are several correlation-based network tools available that can deal with the difficulties of microbiome data, such as the compositionality [14–16]. The potential of correlation-based approaches for uncovering microbial networks has been highlighted in previous research [17].

Whether correlation-based networks represent meaningful ecological structure in microbial communities is however debated. Carr et al. (2019) showed that spurious correlations may occur due to the use of sequencing methods, data transformations and the large number of unmeasured variables [18]. Berry & Widder (2014) and Hirano & Takemoto (2019) assessed the performance of different co-occurrence methods for inferring interaction structure and found that their performance strongly depends on the underlying network properties, like network size and density and the number of samples used to construct the network [13,19]. Apart from the challenges of metagenomic-based abundance data and disagreement between various network tools, here we question whether correlations itself are at all useful to distinguish between different ecological interaction types. Resource competition and metabolic cooperation have been successfully inferred within environmental microbiomes, by linking ecological distribution data to multi-species metabolic models and subsequent verification of putative interactions by means of experimental co-growth analysis [20]. However, host-associated microbiomes often include non-

culturable organisms, without information on nutrient requirements or metabolic function. Likewise, performance of correlation analysis in relation to alternative interaction types in human microbiota is not well understood and deserves further investigation.

Correspondence of correlations with ecological interactions needs to be studied against a known ground truth, which can be achieved by means of simulation. Mathematical models have been used as ground truth in assessment of correlation network techniques before (e.g. [21]), but correlation networks have not been systematically investigated against distinct interaction types in dynamic models. This requires elucidation especially as the 'true' ecological networks governing microbiome dynamics are still unknown. For this purpose, we assessed the performance of correlation-based network reconstruction by simulating abundance data based on the generalized Lotka-Volterra (gLV) model. The gLV model describes the collective dynamics of multiple species by means of an interaction matrix that can modulate different types of interactions [22]. The model is commonly used in microbiome studies for different aims: to simulate microbial communities under various interaction structures [22], to infer interaction structure from time-series data [12], to forecast population dynamics after a perturbation [23], to infer the network topology from steady state samples [24] and to identify the efficiency of intervention protocols in altering the state of a system via the addition or subtraction of microbial species [25]. In ecology, gLV-type models have been questioned for their reliance on pairwise additive interactions, as well as for the strictly linear effects imposed on interspecific interactions. Nonetheless, from the perspective of network inference, it makes sense to first investigate gLV-type models, as their first-order description of ecological dependencies, specified through a pairwise interaction matrix, resembles the objective of correlation analysis and most network models [2].

In addressing how gLV-type interactions can be inferred from cross-sectional data, we mainly focus on the correspondence between the obtained correlation-based networks and the underlying network of ecological interactions. We specifically investigate how inference of microbial interaction types is enabled by interindividual variation in population-dynamic parameters, e.g., species-specific carrying capacities, intrinsic growth rates and strength of interspecific interactions, and how network reconstruction is affected by gLV model assumptions. We highlight several situations where correlations cannot distinguish microbial interaction types, and therefore recommend careful interpretation and validation when inferring networks from cross-sectional abundance data.

## Methods

### Two species Lotka-Volterra model with self-limitation

First, we investigated how interactions between two species of microbial populations are displayed in terms of correlations in abundances in the Lotka-Volterra model. For the sake of convenience, we use the term 'species', although in studies with real microbiome data it is often not possible to characterize the taxonomic abundances at species level and therefore genera or higher taxonomic levels are often used instead.

The two-species Lotka-Volterra model is given by the following set of ordinary differential equations:

$$
\begin{aligned}
\frac{\mathrm{d}N_1}{\mathrm{d}t} &= r_1 N_1 \left( 1 - K_1^{-1} N_1 + \alpha_{12} N_2 \right) \\
\frac{\mathrm{d}N_2}{\mathrm{d}t} &= r_2 N_2 \left( 1 - K_2^{-1} N_2 + \alpha_{21} N_1 \right)
\end{aligned}
\tag{1}
$$

Here, $N_i$ is the abundance of either species 1 or species 2 (with $i = 1$ or $i = 2$). The term $r_i$ is the intrinsic growth rate of each species, here normalized to 1 and 2 per time unit for species 1 and 2 respectively. The effect of each species' abundance on its own growth is defined in terms of the species-specific carrying capacities $K_i$, with $\alpha_{ii} = -K_i^{-1}$ denoting intraspecific competition. We arbitrarily chose the carrying capacity for the first species to be higher than the carrying capacity for the second species ($K_1 = 1.5$; $K_2 = 1.1$), meaning intraspecific competition is less strong for species 1 compared to species 2. Furthermore, $\alpha_{ij}$ ($i = 1, 2$; $j = 1, 2$; $i \neq j$) indicates the interspecific interactions (the effect of one species abundance on the growth of the other species). A positive $\alpha_{ij}$ (e.g., as in the case of mutualism) denotes a positive effect of species $j$ on the growth of species $i$, a negative $\alpha_{ij}$ (e.g., as in the case of competition) means a negative effect of species $j$ on the growth of species $i$ (S1 Fig). We assessed the effect of variation in the interspecific interaction parameters on correlation in equilibrium abundance between both species. For this purpose, the interspecific interaction strengths ($\alpha_{12}$ and $\alpha_{21}$) were drawn randomly from two normal distributions with similar or different mean and similar or different standard deviations ($\sigma_\alpha$). Moreover, we also investigated the situation where $|\alpha_{12}| = |\alpha_{21}|$. Note that it was not possible to achieve stable co-existence for every combination of $\alpha_{12}$ and $\alpha_{21}$. More information on the conditions for co-existence can be found in the supplementary information (S1 Text).

## Generalized host-specific Lotka-Volterra model

Microbial abundance is not only shaped by intra- and interspecific interactions, but also by host characteristics, for example lifestyle, diet and age [26]. Therefore, we investigated the performance of correlation-based network inference of microbial networks for a host-specific version of the gLV model. The host specific gLV model is given by:

$$\frac{\mathrm{d}N_{i,m}}{\mathrm{d}t} = r_{i,m}N_{i,m}\left(1 - K_{i,m}^{-1}N_{i,m} + \sum_{\substack{j=1 \\ j \neq i}}^{s}\alpha_{ij,m}N_{j,m}\right) \qquad (2)$$

Here, $N_{i,m}$ is the abundance of each species $i$ in host $m$, with $i = 1, \ldots, s$ ($s$ being the total number of bacterial species) and $m = 1, \ldots, 300$ (the total number of hosts). The terms $r_{i,m}$ and $K_{i,m}$ are the intrinsic growth rates and the carrying capacities of each species $i$ in host $m$. The carrying capacities are kept separated from the interaction matrix $A$ which only contains interspecific interactions (namely, the pairwise terms $\alpha_{ij}$), facilitating a one-to-one comparison with the correlation matrix.

## Parameterization of the base case simulations

We started with a base case and we added step by step variation to this case. Note that the base-case parametrization does not reflect any particular real-world system. Rather, parameters were chosen in such a way to facilitate computation and promote co-existence between species. Variations to the base-case parameters are shown later on, but also here, findings should be appreciated from a qualitative rather than quantitative viewpoint. In the base case the number of bacteria equals ten. The species-specific growth rate $r_i$ and the species-specific carrying capacity $K_i$ were randomly drawn from uniform distributions respectively U(0.05, 0.1) and U(0, 1). The density of the interaction matrix $A$ in the base case was chosen such that both sparsity of the interaction network and co-existence of the species was promoted in all simulations; in the base case, density was ¼ meaning that three out of four possible interactions were set to zero. Moreover, to ensure co-existence between species in the model we chose

stronger intraspecific interactions than pairwise interspecific interactions. The species-specific parameters $\alpha_{ij}$ were drawn from a Gaussian mixture distribution, as follows. Half of the interactions were drawn from a negative normal distribution: $\alpha_{ij} \sim N(-0.25, 0.1)$; and the other half of the interactions were drawn from a positive normal distribution: $\alpha_{ij} \sim N(0.25, 0.1)$. All interactions were restricted to lie between –0.5 and 0.5, i.e., the normal distributions were truncated at –0.5 and 0.5. The parameters $r_i$, $K_i$ and the interaction matrix $A$ were randomly drawn 1000 times from the aforementioned distributions to obtain 1000 different parameter combinations. Hereafter, host-specific parameters were drawn from log-normal distributions around species-specific parameters, as follows:

$$\begin{cases} \ln(|\alpha_{ij,m}|) \sim N(\ln(|\alpha_{ij}|), \sigma_\alpha) \\ \ln(r_{i,m}) \sim N(\ln(r_i), \sigma_r) \\ \ln(K_{i,m}) \sim N(\ln(K_i), \sigma_K) \end{cases} \qquad (3)$$

Here, $\sigma_\alpha$ denotes the interindividual variability in interspecific interactions among the 300 hosts (with $\sigma_\alpha = 0.25$ in the base case), and $|\alpha_{ij,m}|$ denotes the absolute strength of interaction from species $j$ on the growth of species $i$ for each host $m$. Note that, for the sake of simplicity, the use of log-normal distributions was adopted to induce fold-changes around population means, where both the presence and the sign of interspecific interactions are kept constant across hosts. However, this may be untrue in real microbiota as many microbes can change metabolic pathways and therefore may switch from interaction types and interaction partners. In the base case model, the carrying capacities and growth rates were kept constant across hosts, meaning $\sigma_r$ and $\sigma_K$ were set equal to 0.

The simulation process yielded 300.000 timeseries (300 host specific timeseries for each of the 1000 ten species networks). The running time of the model was chosen such that all species reached their equilibrium abundance. If at least one species did not survive (i.e., when its abundance dropped below 0.001), we rejected the simulation in favor of another randomly drawn parameter set. After sampling the abundances at equilibrium, we added independent and identically distributed noise $v$ to mimic uncertainty in measurements (with $v \sim U(-0.01, 0.01)$ in the base case). This measurement noise can be thought of as representing, for example, sampling errors, environmental contamination, batch effects during sequencing, or annotation errors in reference genomes [27]. Simulations were performed in R (R version 3.6.0; https://www.r-project.org/). The gLV model was solved with the lsoda function from the deSolve package (version 1.24) which uses a FORTRAN ODE solver written by Petzold & Hindmarsh (1995) [28]. R code is available via GitHub (https://github.com/susannepinto/gLV_microbiome.git). A general overview of the base case simulation design is given in Fig 1.

## Variations to the base case model

We studied multiple variations to the base case model. Like the base case simulations, we did 1000 simulations per variation. As a first variation, we added host-specific variability to the species-specific parameters $r_i$ and $K_i$ using Eq 3, with $\sigma_r = 0.25$ and $\sigma_k = 0.25$.

Second, we varied the amount of measurement noise, from $v \sim U(-0.01, 0.01)$ (medium noise in the base case) to $v \sim U(-0.001, 0.001)$ (low noise) and to $v \sim U(-0.1, 0.1)$ (high noise). We also simulated timeseries with a different type of noise, namely varying magnitudes of process noise $W$ (S2 Fig). In contrast to measurement noise, which was added only to the sampled abundances, process noise was added to the gLV model such that within-host population dynamics were perturbed at discrete time intervals $\Delta t$ ($\Delta t = 1$ time unit). The time-varying process noise was drawn from a log-normal distribution to prevent the abundances from

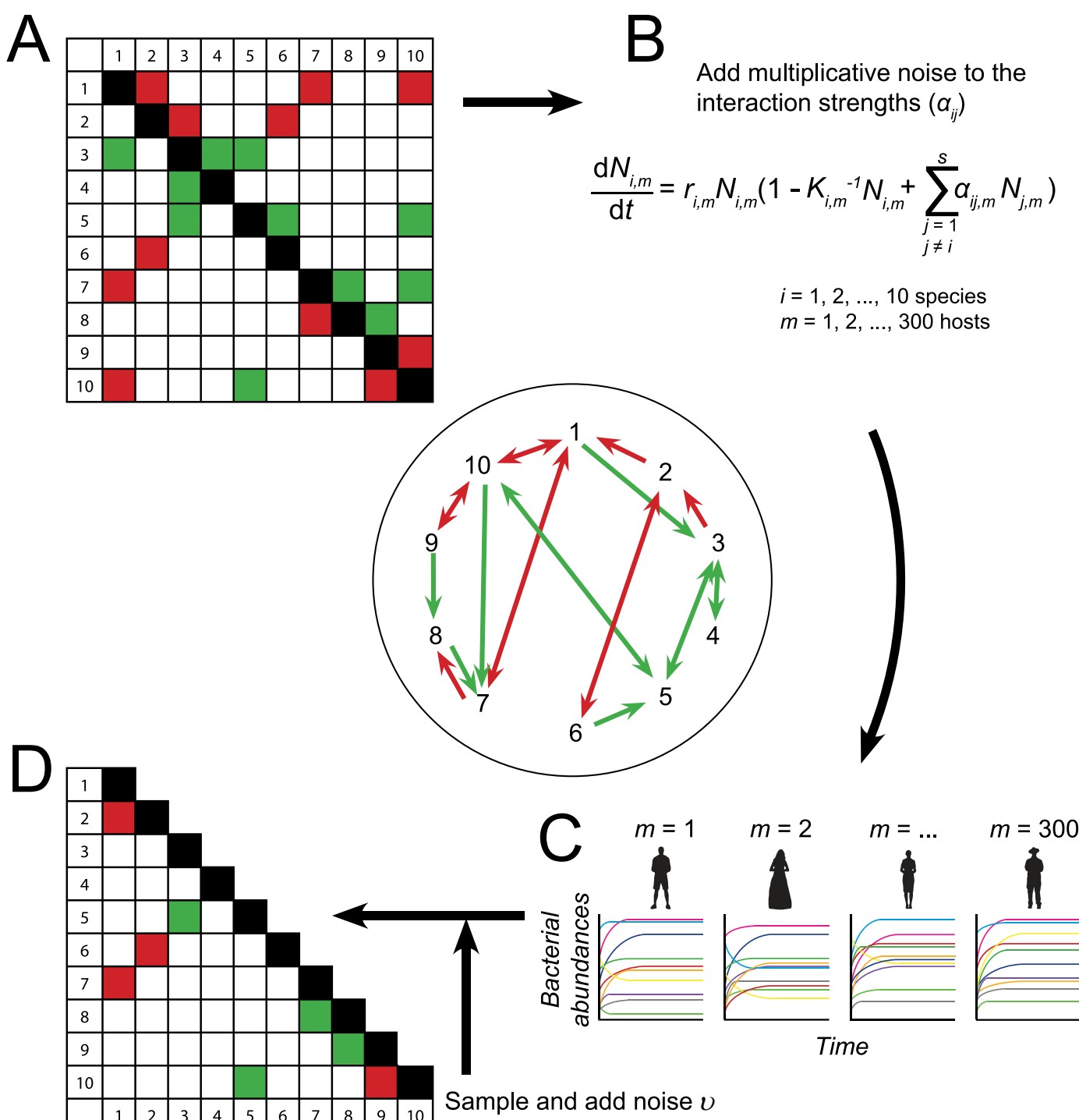

**Fig 1. Representation of the workflow.** In an interaction network singular green and red arrows represent a commensalistic interaction and an amensalistic interaction respectively, whereas double green arrows represent mutualism and double red arrows competition. A green and red arrow signifies an exploitative interaction. See S1 Fig for more details. (A) A random interaction matrix $i$. This interaction matrix is implemented in the gLV model (B) together with the intrinsic growth rates and carrying capacities of the species. (C) All timeseries are (slightly) different due to the variation in the interaction strengths. (D) The partial correlations are calculated from the abundances per species sampled from the 300 different hosts at equilibrium. Only the significant correlations and the lower part of the matrix are used for the comparison with the original interaction matrix $i$. Variations to the workflow were studied by adding for example a perturbation or process noise.

dropping below zero, i.e. $\Delta W_i = \ln(N_{i,m(\Delta t)}) - \ln(N_{i,m(t)}) \sim N(\ln(N_{i,t}), \sigma_W)$ (with $\sigma_W \sim N(0, 1)$ for high process noise and $\sigma_W \sim N(0, 0.1)$ for low process noise).

Further, we simulated data with interaction strengths drawn from a uniform ($\alpha_{ij} \sim U(-0.5, 0.5)$) or unimodal ($\alpha_{ij} \sim N(0, 0.15)$) distribution. As in the base case, the interaction strengths were restricted to lie between –0.5 and 0.5 (S3 Fig).

We also analyzed three different structures of microbial networks. First, we increased the number of species $s$ to 30. To promote co-existence, we also reduced the density of the interaction matrix to 1/6. Secondly, we simulated a network based on a producer consumer relation between the species (S4 Fig). Instead of random interaction networks (S4A Fig), the producer-consumer networks are based on a cross-feeding structure between producers and consumers (with equal numbers of producers and consumers) (S4B Fig). Producers excrete metabolites which are consumed by the consumers. Because consumers remove the 'waste' from the producers, the presence of a consumer can also be beneficial for the producers. Therefore, between producers and consumers positive interactions are more likely to occur than negative interactions. On this purpose, we drew the consumer-producer interactions from the positive side of the Gaussian mixture distribution ($\alpha_{ij} \sim N(0.25, 0.1)$). In contrast, among producers and consumers themselves, the interactions are predominantly negative as these species are more likely to compete for similar resources. On this purpose, we drew the interactions among producers and among consumers from the negative side of the Gaussian mixture distribution ($\alpha_{ij} \sim N(-0.25, 0.1)$). Thirdly, we simulated a microbial network with interaction hubs, i.e. a network containing species with unusually high numbers of ecological interactions compared to other species in the network (S4 Fig) [29]. Hub-species networks were created according to the Barabási-Albert model [30] and implemented with the barabasi.game function from the igraph package (version 1.2.11). In the network-generating algorithm, interactions are distributed according to a mechanism of preferential attachment. Thus, species with interactions obtain a higher chance of getting more interactions, resulting in a few 'hub-species' with many interactions. We constructed two scale-free directed graphs (with power = 2), denoting "incoming" and "outgoing" interactions, and combined these to obtain a bidirected graph. Density was kept similar to the base case model (1/4).

Next, we also investigated how network inference is affected by sample size by considering a scenario with 3000 instead of 300 hosts. We did this for the base-case model with random interaction networks, as well as for the producer-consumer and hub-species networks described above.

Last, we investigated the effect of a perturbation on the performance of network inference. The populations were perturbed after 175 time units, with a perturbation that lasted for 50 time units. The perturbation was modelled by taking a new set of random carrying capacities per species per sample. Due to the simulated perturbation, the equilibrium distribution shifted. After the perturbation, the species grew back to their original equilibrium. Sampling occurred before, during or after the perturbation.

## Assessment of correlation-based network inference

With the simulated data at hand, we created a dataset with the abundances of the model species sampled at equilibrium for each host $m$. After adding measurement noise to the data, we inferred the correlations between species by calculating the partial Pearson correlation coefficients $\rho$ between all abundances $N_i$ across the $m$ different hosts (Fig 1). We did not use plain correlations, because partial correlations have the advantage of controlling for confounding interactions (e.g. interactions between bacterial species affecting the abundance of a third species) [31]. Agreement between the partial correlation matrix and the interaction matrix $A$ from the gLV model was assessed qualitatively, i.e., we only considered whether significant

**Table 1. The confusion matrix as used in this study.** The inferred partial correlation coefficient $\rho$ (from the lower part of the partial correlation matrix) must have the same sign as one of the interactions in the interaction matrix $A$ to be considered as a true positive finding in base case analysis.

| Interaction in the $A$ matrix from the model | | Inferred partial correlation | | |
|---|---|---|---|---|
| | | Negative* | Not significant | Positive* |
| No interaction | 0, 0 | false positive | true negative | false positive |
| Mutualism | +, + | false positive | false negative | true positive |
| Competition | –, – | true positive | false negative | false positive |
| Commensalism | +, 0 \| 0, + | false positive | false negative | true positive |
| Amensalism | –, 0 \| 0, – | true positive | false negative | false positive |
| Exploitative interaction | +,–\|–, + | true positive | false negative | true positive |

* Only significant partial correlations (with $p < 0.05$) are considered after correction for multiple testing with Benjamini-Hochberg procedure.

entries in the partial correlation matrix agreed with the interaction matrix in terms of non-zero entries with the correct sign. We used the Benjamini-Hochberg procedure to control for the expected proportion of 'false discoveries' after calculating partial correlations between each pair of species [32]. The results (true positives, true negatives, false positives and false negatives) were stored in a confusion matrix (Table 1). Because a correlation matrix is symmetric and an interaction matrix $A$ is not, we only used half of the partial correlation matrix (Fig 1D) to construct the confusion matrix. For a correctly classified interaction, either one or both interactions in the upper and lower part of the $A$ matrix must have the same sign as in the lower part of the partial correlation matrix. This can produce a bias, because asymmetric interactions can result in a true positive result for correspondence of the correlation coefficient ($\rho$) with either interaction. For example, for exploitative interactions, both negative and positive correlations are classified as true positive results. Therefore, we tested the effect of this bias on the success of network inference by specifying the intended sign in correlation analysis, as the sign of the strongest interaction in each pair of species. Hence, for an exploitative interaction, only a positive or a negative correlation is correct, depending on the weights of the asymmetric interactions. Secondly, we also tested the effect of this bias on the success of network inference by setting the rule that the sign of both interactions must be matched by the inferred correlation coefficient. Hence, only mutualism and competition can be inferred correctly, as amensalism, commensalism and exploitative interactions are asymmetric.

Performance of network inference was evaluated with precision and recall and a combination of both measures, called the F1-score [33]. The precision is the fraction of correctly classified interactions among the total number of significantly predicted interactions (i.e., significant partial correlations) and the recall is the fraction of correctly classified interactions among the total number of non-zero interactions in the interaction matrix $A$. The F1-score (on a scale from 0 (no agreement) to 1 (perfect agreement)) is obtained as the harmonic mean of precision and recall, weighted equally, as given in the following equation:

$$F_1 = 2 \cdot \frac{precision \cdot recall}{precision + recall} \tag{4}$$

## Results

### Inference of asymmetric and symmetric interactions in a two-species system

Correlations in abundances of the species in a two-species Lotka-Volterra model are shaped by the type of interaction involved. Fig 2 shows scatterplots of the abundances of two bacterial

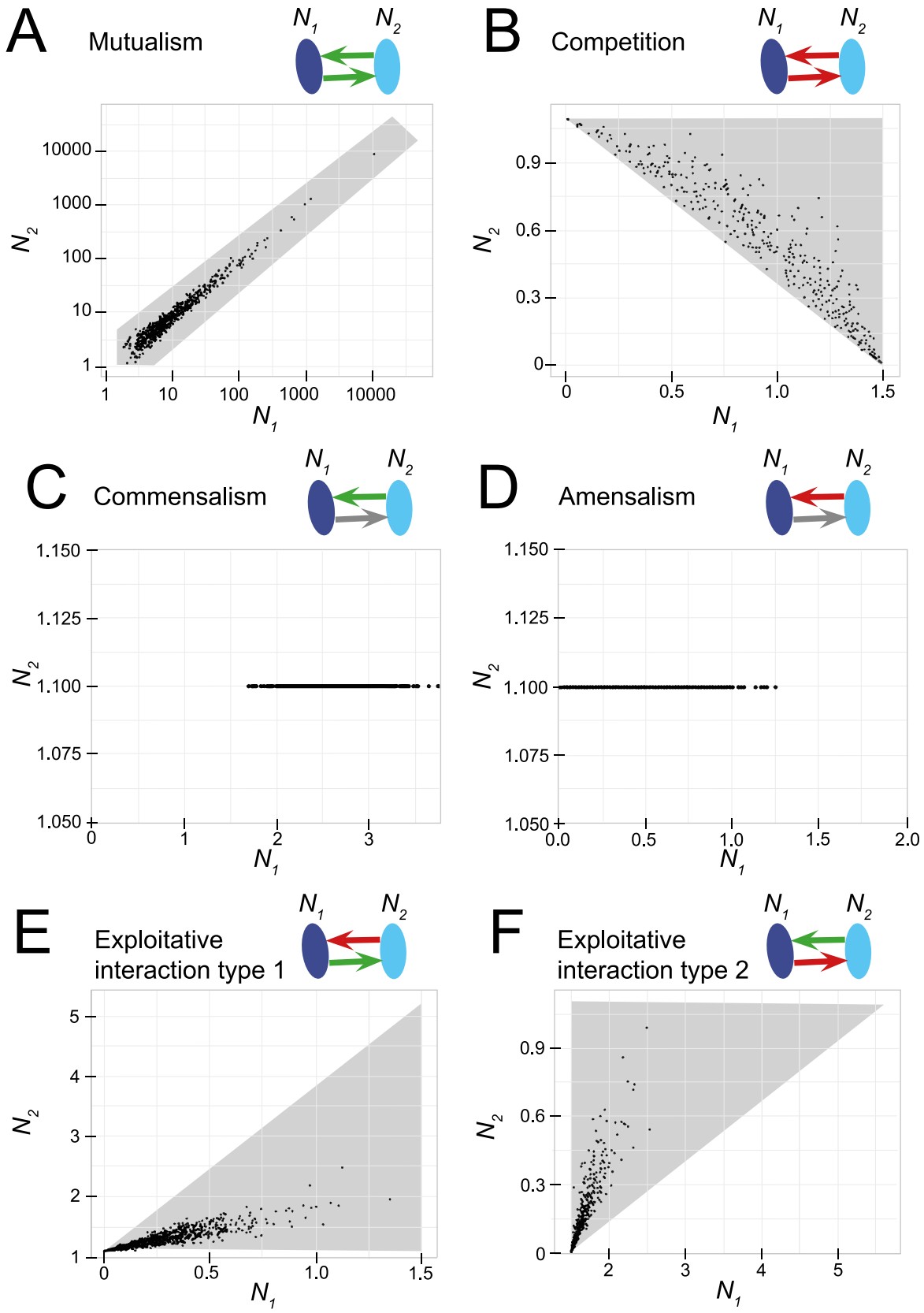

**Fig 2. Scatter plots between the abundances of two bacterial species for different interaction mechanisms: (A) mutualism, (B) competition, (C) commensalism, (D) amensalism and (E, F) exploitative interactions.** The abundances of the two species $N_1$ and $N_2$ at equilibrium are shown as scatterplots and have been obtained by running the two-species Lotka-Volterra model, with $K_1 = 1.5$; $K_2 = 1.1$; $r_1 = 1$; $r_2 = 2$ and $\alpha_{ij}$ drawn randomly from normal distributions with identical means and standard deviations ($\alpha_{12} \sim N(|0.7|, 0.2)$, $\alpha_{21} \sim N(|0.7|, 0.2)$). In the case of commensalism and amensalism: $\alpha_{12} \sim N(|0.7|, 0.2)$ and $\alpha_{21} = 0$. The two species can co-exist under certain combinations of $\alpha_{ij}$ (S1 Text). The grey polygon indicates the area where co-existence is possible. Note that the axes have different ranges in each subplot. Because the two species have different carrying capacities, the two situations of exploitative interactions are different. i.e., in case of exploitative interaction type 1: species 1 is exploited by species 2 and in case of exploitative interaction type 2: species 2 is exploited by species 1.

species for different interaction mechanisms over a range of different combinations of $\alpha_{12}$ and $\alpha_{21}$. Mutualistic interactions clearly yielded a positive correlation in abundance between the two species involved (Figs 2A and S5). Competitive interactions generally yielded negative correlations (Figs 2B and S5). However, under perfectly symmetric competition (when $\alpha_{12} = \alpha_{21}$) we did find a positive correlation depending on interaction strength and carrying capacities of the species involved (S5D Fig, second panel). In the situation where one of the two species does not experience any benefits or limitations in growth from the other species, as is the case with commensalism and amensalism (i.e. $\alpha_{12} = 0$ or $\alpha_{21} = 0$), correlations are zero because one of the species will grow to its carrying capacity irrespective the abundance of the other species (Fig 2C and 2D).

Correlations under exploitative interactions among bacteria, benefitting one but harming the other species, generally yielded positive correlations (Figs 2E, 2F and S5), but negative correlations were also found. This happened when the exploitative benefit was of equal magnitude as the harm done to the other species (S5D Fig), or of similar mean magnitude but with more variation (e.g. species 1 is exploited by species 2; $-\alpha_{12} = \alpha_{21}$ and $\sigma_{\alpha12} << \sigma_{\alpha21}$ (exploitative interaction type 1) or species 2 is exploited by species 1; $\alpha_{12} = -\alpha_{21}$ and $\sigma_{\alpha21} << \sigma_{\alpha12}$ (exploitative interaction type 2) (S5B Fig). However, if the exploitative benefit outweighs the harm done to the other species, exploitative interactions will generally yield positive correlations. It should also be noted that the two species were not exchangeable, because species 1 was given a weaker intraspecific interaction strength than species 2. Thus, in the absence of interspecific interactions, species 1 can reach a higher abundance at equilibrium. This means that, for the same interspecific interaction strength, the species with the higher carrying capacity exerts a stronger (negative) effect on the growth of the other species.

## Network inference under various interaction types

Here we used the base case model to assess the success rate of recovering a particular interaction type between pairs of species: amensalism, commensalism, exploitative interactions, mutualism and competition (S1 Fig). Fig 3A shows that correlations were more often found in mutualistic and competitive interactions, where interacting species experience the same qualitative effects from each other, than in amensalistic and commensalistic interactions, where only one species experiences an effect from the presence of another species. For exploitative interactions among bacteria, either a positive or negative correlation coefficient $\rho$ could be found, with a success rate comparable to amensalistic and commensalistic interactions. Contrary to the results that included symmetric interactions, there was no difference between the successful inference of positive interactions over negative interactions in any interaction type (Fig 3B). For all interaction types, the sign of the significant correlation coefficient $\rho$ found, mostly agreed with the sign of the type of the interaction (Fig 3A and 3B). However, with the inferred correlations neither the type nor direction of the original interaction could be recovered.

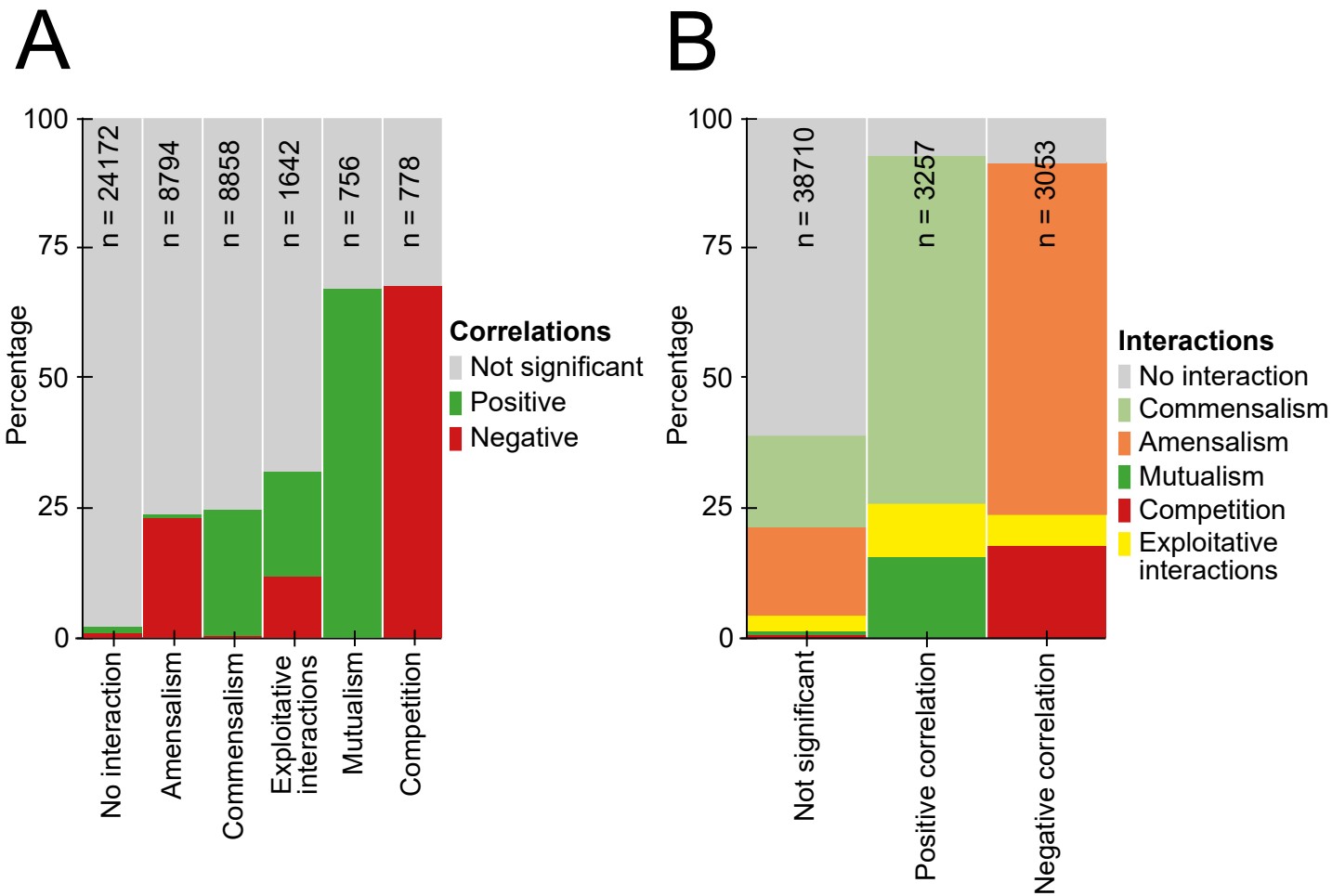

**Fig 3. The percentage of significant partial correlations (with sign matching interaction in either direction), as recovered from the base case model.** (A) For different types of pairwise interactions and (B) for the different correlations.

### Network inference under various sources of process variability

Next, we investigated how correct network inference was affected by several variations to the base case model (Fig 4 and S1 Table). In all cases considered, interactions were recovered with precision exceeding recall. This means that the likelihood of missing an interaction (i.e., 1 – recall) was higher than the likelihood of finding a false interaction (i.e., 1 – precision), illustrating the effect of false discovery rate control.

Partial correlations corresponded to non-zero entries in the interaction matrix only when interindividual variation existed in the interaction parameters ($\alpha_{ij}$) and/or carrying capacities ($K_i$) (Fig 4A and 4B). These parameters directly influence microbial abundance patterns, as interspecific interactions and carrying capacities determine the equilibrium of the gLV model. The intrinsic growth rate only determines the speed at which species reach their equilibrium, and this parameter is not informative for the equilibrium abundances. In fact, performance under interindividual variation in growth rates was just as bad as the performance under pure measurement noise with no variation in model parameters (Fig 4B).

Performance of correlation-based network inference was robust to measurement noise, if measurement noise was small compared to interindividual variation in process parameters

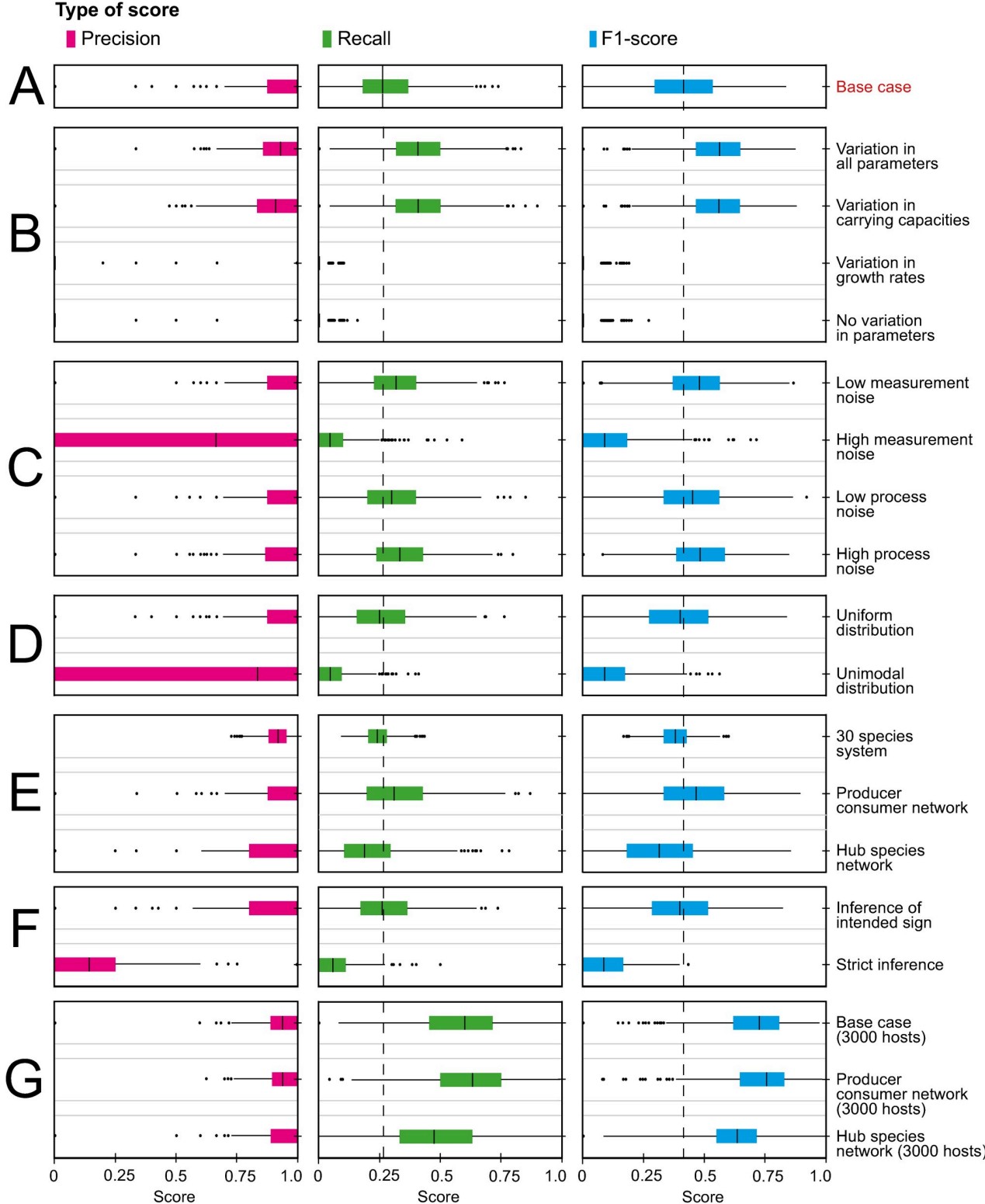

**Fig 4. Inference under various sources of process variability.** For the different scenario's we show the precision, recall and the F1-score. (A) The base case model. (B) Host-specific variation in the carrying capacities and intrinsic growth rates. (C) Decreased and increased amount of measurement noise ($v$) and the effect of process noise ($W$) (S2 Fig). (D) Interaction strengths drawn from a uniform and unimodal distribution (S3 Fig). (E) The results for a 30 species system, a network based on a producer-consumer structure and a network with hub interactions (S4 Fig). (F) The effect of network inference when specifying the intended sign in correlation analysis, as the sign of the strongest interaction in each pair of species, or

by setting the rule that the sign of both interactions must be matched by the inferred correlation coefficient (strict inference). (G) Three scenarios with 3000 hosts, for the base-case with random interaction networks as well as for the scenarios with structured (i.e. producer-consumer and hub-species) networks. Network inference was assessed by the F1-score, which measures agreement between the interaction matrix in the gLV model and the inferred partial correlation matrix on a scale from 0 (no agreement) to 1 (perfect agreement) (according to the rules of Table 1). The dashed line indicates the median result from the base case model. The bars of the boxplots indicate the variability of the data outside the middle 50% (i.e., the lower 25% of scores and the upper 25% of scores).

(Fig 4C). When measurement noise became of the same magnitude as the variation in inter-specific interactions, the F1-score deteriorated, and it was no longer possible to use correlations as a proxy for interactions (Fig 4C). We also checked whether adding process noise would affect the inference. We did observe a significant improvement of the inference from a model with process noise relative to only measurement noise (Fig 4C and S1 Table).

Hereafter, we investigated the effect of drawing the interaction strengths from different types of distributions (Figs 4D and S3). We did not observe a difference between the success rate of network inference under a Gaussian mixture distribution or uniform distribution, which were conditioned to have similar variances (S1 Table). However, successful inference deteriorates with reduced interaction strength, as success rates were better under a Gaussian mixture distribution or uniform distribution, as compared to a unimodal distribution around zero (with smaller variance) (Fig 4D). The weaker interactions have a smaller effect on equilibrium abundances of other species, which makes them harder to detect with correlation analysis.

Fig 4E shows the results for different network types. Increasing the number of species from 10 to 30 had a significant negative effect on the success of the inference (S1 Table), which was mainly due to reduced precision. Conversely, F1-scores were improved as compared to the base-case when assuming a producer-consumer based network (S4 Fig and S1 Table), on account of an improved recall. The inference in a network with interaction hubs (as explained in S4 Fig) was significantly worse than in a random network, which could be attributed to a somewhat reduced recall.

Note that problems may arise with asymmetric relationships. When using the rule that pairwise correlations should match the strongest interaction between both species involved as the intended sign, we found only a slight non-significant reduction in F1-score as compared to the base case scenario (Fig 4F and S1 Table). Thus, pairwise interactions wherein the net effect on population growth is positive or negative are mostly picked up as such in correlation analysis. However, under the rule that mutual interactions must both be reflected in the sign of the correlations, asymmetric interactions cannot be recovered as correlations are symmetric. We indeed found much lower F1-scores when detection of asymmetric interactions was no longer considered as a true positive result after inferring a significant correlation coefficient $\rho$ (either positive or negative) (Fig 4F).

Finally, we verified that network inference improved with increasing sample size. This applied to models with random as well as structured interactions networks (Fig 4G). In the base case, precision was somewhat reduced at increased sample size notwithstanding Benjamini-Hochberg control. However, this was compensated by substantially improved recall, resulting in significantly increased F1-scores. Interestingly, precision stayed more or less constant at increased sample size in producer-consumer and hub-species networks, whereas recall improved but remained somewhat behind that of random networks.

## Network inference under non-equilibrium conditions

Fig 5 shows that the equilibrium assumption is not necessary for successful correlation-based network inference. In fact, our results even suggest that a perturbation can positively affect the

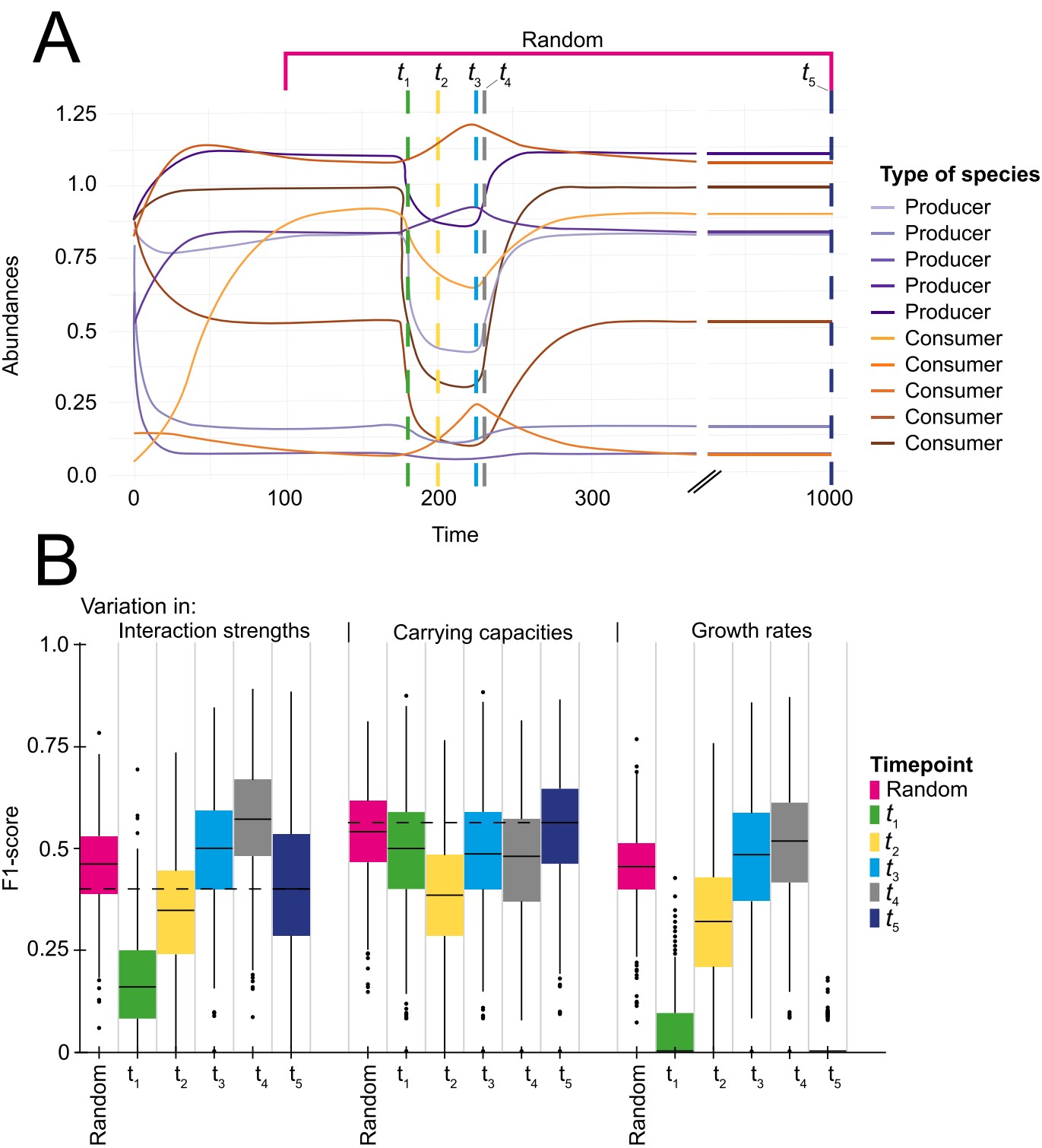

**Fig 5. The effect of a perturbation on correlation-based network inference.** (A) Example of a timeseries. Dashed lines represent sampling timepoints. Sampling was performed during the perturbation ($t_1$ = green, $t_2$ = yellow, $t_3$ = blue and $t_4$ = grey) and at equilibrium ($t_5$ = dark blue). Alternatively, sampling was performed randomly between $t = 100$ and $t = 1000$ (random = pink). (B) Results (F1-scores) of network inference for sampling at various timepoints. After a perturbation all species grow back to their original equilibrium. The bars of the boxplots indicate the variability outside the middle 50% (i.e., the lower 25% of scores and the upper 25% of scores). Dashed lines represent median results of sampling during equilibrium.

performance of network inference. Variation in the growth rates becomes significantly informative outside the equilibrium (S2 Table). Also, variation in the interactions becomes even more informative when the population is still growing towards the equilibrium. Network inference is impaired only right after the start of a perturbation, when the population is still far from a new equilibrium, unless the interindividual variation is in the carrying capacities (Fig 5B). We also assessed the success of correlation-based inference when the sampling occurred randomly in time in relation to the perturbation. We found that the F1-score resembled an average of F1-scores across various sampling timepoints.

## Discussion

Correlation-based network inference has been used in many studies and for many different types of human and environmental microbial communities [31]. The reliability of the results with regards to true ecological dependencies has been criticized, to the extent that correlation analysis has been suggested to almost never reveal anything substantive about the biotic relationships between bacteria [18]. However, the theoretical basis that enables ecological interactions to be inferred from cross-sectional abundance data remains poorly understood. Most of the previous research has focused on the reconstructed network properties or the difficulties pertaining to metagenomics-based abundance patterns, e.g., the compositionality of the data and the high proportion of zeros [18,31,34]. While these difficulties are pervasive and merit further consideration, here, we question whether correlations are at all useful in distinguishing different interaction types in microbial networks.

We demonstrated multiple pitfalls when using correlation-based methods for inferring interactions. Some of those pitfalls are well known, as they relate to the inherent symmetry of correlation-based metrics and the frequent asymmetry of ecological interactions [18]. As a result, asymmetric interaction types (commensalism, amensalism and exploitative interactions) cannot be recovered with indication of the direction of interaction, which agrees with prior work done by Weiss et al. (2016) [21]. Symmetric interaction types, where species involved affect each other's growth in a qualitatively similar way (competition, mutualism) can be recovered, although competitive interactions may also result in positive correlations, albeit in very rare cases where species have identical competitive strength. Likewise, we found that exploitative interactions generally induce positive correlations, especially in the likely circumstance where the exploitative benefit outweighs the harm to the exploited species. These findings might explain why empirical correlation-based networks have a relative shortage of negative correlations [20,34,35]. It remains to be investigated whether the high frequency of positive edges in reconstructed networks is caused by methodologic limitations or whether the interspecific interactions in host-associated microbiota are primarily mutualistic [36–39].

Still, as illustrated by our analysis, correlations in microbial abundance across independently sampled hosts can be indicative for underlying ecological interactions under host-specific variation in microbial population dynamics. That is, if microbial groups of interest are omnipresent and their interactions are appropriately captured by generalized Lotka-Volterra (gLV) dynamics, the variation in population abundances should be driven by interindividual variability in population-dynamic parameters. In the context of the gLV model, the informative parameters are primarily related to intrinsic growth rates, carrying capacities and strength of between-species interactions of microbial groups considered. A change in species abundances can be informative for the interactions among those species, as was also previously shown by Stone and Roberts (1991) [40]. It remains to be determined how much variability across individual hosts is driven by external forcing and by gradual differences in process-related parameters relative to measurement noise. On one hand, it is well known that microbes

adapt to host-specific environments, shaped by diet, lifestyle, hormonal regulation, immune system, etcetera [26]. As an example, increased abundance of a particular bacterial species at increased glucose intake levels might be reflective of increased resource availability (affecting carrying capacity and growth rate) or superior competitive strength (affecting interactions with other species) [6]. On the other hand, environmental drivers of bacterial growth can operate over different spatial and temporal scales and correlations in abundance can be reflective of shared environmental niches that have no meaning in terms of direct biotic interactions [1].

Therefore, a correlation between the abundance of two species does not imply that those species are interacting [41]. Many of the detected correlations may be caused by shared environmental preferences rather than species interactions [42]. Such kind of environmental filtering can mask putative between-species interactions as well as induce spurious correlations [18]. Also, co-occurring species may appear to be dependent on each other, while their co-occurrence can be explained by them actually sharing a similar dependency on a third party– so that co-occurrence, and hence apparent dependencies drawn from that, may also be explained by higher-order interactions [43]. Berry and Widder (2014) claimed that network interpretation is only possible if samples are derived from similar environments [13]. Our analysis suggests that network inference partially depends on a degree of heterogeneity in population-dynamic parameters. If differences in bacterial abundances between hosts are mainly due to measurement noise, their correlations are not informative of underlying interactions. In our simulations, with relative standard deviation in process-related parameters between hosts of about 25%, inference performed well as long as measurement noise had coefficients of variation well below 10% of mean bacterial abundances. Strikingly, the inference of interactions was even improved when process noise was added. More research is needed to delineate the extent to which correlation analyses can be confounded by latent environmental drivers of microbial population dynamics, and how strongly one should condition on environmental or host homogeneity.

Our results have been obtained by using the gLV model. While the gLV model has been very popular in microbiome research because of its manageability, it has several drawbacks. In ecology, the gLV model has been criticized for the absence of trophic levels within the model [44]. This is in contrast to most classical ecological (e.g. plant-herbivore or predator-prey) systems, where direct consumption and predation offer more opportunity for top-down regulation, possibly obscuring interactions in co-occurrence patterns [45]. But trophic levels are probably not so relevant in the human microbiome as bacteria mainly interact with each other through excreted metabolites [2]. Furthermore, the interactions between bacteria might be much more complex than the additive and pairwise interactions that the gLV model assumes. Momeni et al (2017) claimed that pairwise modeling will often fail to predict microbial dynamics, as many interactions occur through chemical production pathways (such as cross-feeding and nutrient competition) involving more than two species [46]. Correlation analysis fails to capture the resulting higher-order interactions, for which more advanced techniques, e.g. graphical models [47], might be more appropriate. It is unclear, how well directed links predicted by these methods recover true ecological interaction types. Often, they require more prior knowledge of the network of microbial interactions, time series or more fine-grained data on the pathways of interaction. Moreover, microbial networks can be bi-directed and cyclic [20], which poses problems for inference of directionality and type of interactions from mere cross-sectional data. More classical methods of separating direct from indirect interactions, e.g. path analysis [48], rely on testing of specific alternative causal hypothesis, which can only be considered as a next step in network inference. To shed more light on causal pathways, there is a need in microbial ecology for models that can describe the full set of metabolite concentrations, metabolic fluxes and species abundances within a community [49]. Based on

metabolic modelling, Freilich et al. (2011) concluded that cooperative interactions are relatively rare among free-living bacteria and, if present, are often unidirectional. Machado et al. (2021) suggested that mutualistic interactions are much more common among host-associated bacteria, that often form highly cooperative communities and have smaller genomes and fewer metabolic genes compared to other species. Cooperative communities are resilient to nutrient change and adaptable to a wide variety of different environments, including the human body [20,43]. Metabolic modeling is still challenging and heavily based on a priori assumptions, but is also a rapidly developing field that may prove useful for computational validation of correlation-based interaction networks [50].

In addition, the gLV model disregards important biological processes, such as adaptation (for instance, switching of mutualistic partners due to for example horizonal gene transfer [51]), that may affect the topology of ecological networks, rather than the strength of ecological interactions in a network. Furthermore, the gLV model displays dynamics that are characterized by strong equilibrium attractors. Many studies have shown the occurrence of complex dynamics as alternative stable states [52], oscillations and chaos in experimental [53–55], but also in field studies [56], with ecological communities. Whether this also applies to the bacterial communities inhabiting the human body is still unknown, due to the paucity of long-term human microbiome studies. However, a study among a thousand western individuals has suggested the existence of tipping elements in the intestinal microbiome [57] indicating the possible presence of alternative attractors in the dynamics of gut microbiome communities [58,59].

As a general critique, the use of simulated data based on gLV dynamics raises the question to what extent the necessary model assumptions (and therefore the results) are representative for the human microbiome. Of course, real data are much more complex than simulated data. To reiterate, our base-case parametrization does not reflect any particular real-world system, and findings should be appreciated from a qualitative rather than quantitative viewpoint. Even so, while models can only serve as very crude approximations, the main features of model-based analysis might still hold, as demonstrated by Freilich et al. (2018) [42]. They compared a well-resolved, empirically defined interaction network of species in the rocky intertidal zone in Central Chili to a reconstructed network based on the co-occurrence of those species. There are similarities in their findings to our results. For example, they found that weak interactions are missed more often than interactions above a certain threshold. They also concluded that the ability to correctly detect a true link varies across different interaction types, and that positive interactions are better detected than negative interactions. Interestingly, in line with our results, they also found that negative interactions are misclassified as positive interactions more often than vice versa.

In our simulation studies, the chance of finding false interactions was well under control by using partial correlations with adjustment for multiple testing. It should be noted that application of correlation-based network reconstruction to real-world high-throughput microbial abundance data typically requires additional constraints for control of false discovery rates. Real-world microbiome data have some specific challenges which may negatively affect the success of correlation-based network inference. The compositionality of the data, the diversity of species (with many rare species) and the density of interactions make these networks harder to predict and apparent correlations more likely to appear [14,19]. Various correlation-based methods, often free of charge and stored in pre-programmed packages are available to handle these challenges. However, Weiss et al. (2016) showed that with the same data, there is much disagreement between the inferred networks generated by different tools [21]. Thus, even if correlations are a useful proxy of microbial interactions, performance of network inference in high-dimensional settings will also strongly depend on the specific network modelling approach taken.

To summarize, correlation-based methods are particularly insensitive for the detection of asymmetric interactions (such as exploitative interactions, amensalism or commensalism), as direction of interaction cannot be recovered from co-occurrence data. Still, they may perform well when applied to networks that are dominated by mutualistic and competitive interactions, as in producer-consumer systems. Applicability of correlation-based network inference to readily available microbiome data thus depends on the type of interactions that govern microbiome dynamics, which likely depends on each application. To conclude, our study suggests that hypotheses about microbial interactions, generated with correlation-based methods, should be questioned with domain-specific knowledge. We highlight again the careful interpretation and validation that is required.

## Supporting information

**S1 Text. Co-existence in a two-species Lotka-Volterra model with self-limitation.** Table A. Conditions for stable co-existence in the two-species Lotka-Volterra model. Fig A. Zero-growth isoclines ("null-clines") in the two-species Lotka-Volterra model.
(PDF)

**S1 Fig. Cartoon illustrating the different interaction mechanisms.**
(PDF)

**S2 Fig. The effect of process noise ($W$) on the within host population dynamics.**
(PDF)

**S3 Fig. Distributions of interaction strengths in three different scenarios.**
(PDF)

**S4 Fig. Network structures used in the different case studies.**
(PDF)

**S5 Fig. The effect of $\alpha_{ij}$ on the correlations between the abundances of two bacterial species for different interactions mechanisms.**
(PDF)

**S1 Table. Mann-Whitney U test results for the F1-scores of the base case model and for the F1-scores of the model with different sources of process variability.**
(PDF)

**S2 Table. Mann-Whitney U test results for the F1-scores of the samples taken during equilibrium ($t_5$ in Fig 5) and for the F1-scores of the samples taken outside equilibrium.**
(PDF)

## Author Contributions

**Conceptualization:** Susanne Pinto, Elisa Benincà, Egbert H. van Nes, Marten Scheffer, Johannes A. Bogaards.

**Data curation:** Susanne Pinto, Elisa Benincà, Johannes A. Bogaards.

**Formal analysis:** Susanne Pinto, Elisa Benincà, Egbert H. van Nes, Johannes A. Bogaards.

**Funding acquisition:** Elisa Benincà, Johannes A. Bogaards.

**Investigation:** Susanne Pinto, Elisa Benincà, Egbert H. van Nes, Marten Scheffer, Johannes A. Bogaards.

**Methodology:** Susanne Pinto, Elisa Benincà, Egbert H. van Nes, Marten Scheffer, Johannes A. Bogaards.

**Project administration:** Susanne Pinto, Elisa Benincà, Johannes A. Bogaards.

**Resources:** Susanne Pinto, Elisa Benincà, Egbert H. van Nes, Marten Scheffer, Johannes A. Bogaards.

**Software:** Susanne Pinto, Elisa Benincà, Johannes A. Bogaards.

**Supervision:** Elisa Benincà, Johannes A. Bogaards.

**Validation:** Susanne Pinto, Elisa Benincà, Egbert H. van Nes, Johannes A. Bogaards.

**Visualization:** Susanne Pinto, Elisa Benincà, Johannes A. Bogaards.

**Writing – original draft:** Susanne Pinto, Elisa Benincà, Egbert H. van Nes, Marten Scheffer, Johannes A. Bogaards.

**Writing – review & editing:** Susanne Pinto, Elisa Benincà, Egbert H. van Nes, Marten Scheffer, Johannes A. Bogaards.

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
