## [Decision Letter · Decision Letter 0]

24 Feb 2022

Dear Dr. Pinto,

Thank you very much for submitting your manuscript "Species abundance correlations cannot distinguish interaction types in microbial networks" for consideration at PLOS Computational Biology.

As with all papers reviewed by the journal, your manuscript was reviewed by members of the editorial board and by several independent reviewers. In light of the reviews (below this email), we would like to invite the resubmission of a significantly-revised version that takes into account the reviewers' comments.

We cannot make any decision about publication until we have seen the revised manuscript and your response to the reviewers' comments. Your revised manuscript is also likely to be sent to reviewers for further evaluation.

Sincerely,

Kiran Raosaheb Patil, Ph.D.

Deputy Editor

PLOS Computational Biology

Reviewer's Responses to Questions

**Comments to the Authors:**

Reviewer #1: In this manuscript, the authors approach the question of whether cross-sectional sampling of microbial communities, followed by correlation analysis of quantitative abundances, can reveal insights into the types of ecological interactions occurring between microbes in a community. The authors test this by performing Lotka-Volterra simulations under known interaction topologies, and varying the parameters and/or noise-levels during each run. They then perform partial correlation analysis across multiple snapshots that have resulted from the simulations, testing whether the original input interactions can be retrieved.

Overall, the study is dealing with a relevant problem, and it is technically well executed. However, I doubt whether its findings are all that surprising/novel. Most scientists would doubt that detailed mechanistic (ecological) insights could ever be drawn from cross-sectional correlation analysis alone, and I am not aware of studies that conclude about fine differences between directionalities and types of interactions from mere correlation data. More importantly, there are more advanced techniques such as Structural Equation Models or Bayesian Networks, which attempt to infer causality from cross-sectional community data, but are not tested by the authors.

Furthermore, one of their novel findings (namely that parasitic interactions can manifest in both positive as well as negative correlations) is perhaps somewhat unrealistic: negative correlations are only observed in setups where the positive effect of the host on the parasite is much smaller than the negative effect of the parasite on the host (cf. line 329 and following). This appears to be biologically unrealistic; most parasites have evolved to avoid overly straining the host … the above setup is somewhat difficult to perceive as biologically meaningful/stable.

Also, given that generic interaction types (positive and negative) are by-and-large well detected (Fig. 2 and 3A), the title of the study is perhaps unnecessarily pessimistic. A bit more nuance would perhaps be more helpful to the reader.

The parameter and design choices during the modelling are fairly arbitrary: the interaction networks are small and have an entirely random topology – meaning that important features such as interaction-hubs and -modules are not considered. The authors rarely motivate their design and parameter choices, which makes the interpretation of their observations difficult. The level of noise in the “high” noise settings seems large – in my opinion it is not clear whether such levels of measurement noise prevail in practice.

Their "strict inference" analysis (Fig. 4F) would be more fair, if in cases where the true interaction weights are asymmetric, but the correlation analysis yields the “sign” of the interaction with the larger weight, this would be counted as “true positive” predictions. As stated now, the task cannot be solved, since the intended sign is not specified.

In addition to the concerns above, there are a number of minor points as below:

- “only a few samples at a time” repeatedly used, but misleading, since most studies have dozens or hundreds of samples (not to mention large scale projects with thousands)

- Figure reference in line 329 should probably be Fig. 2E/2F, since 2C/2D don’t show parasitism

- L 407: the authors use the term significant without statistically quantifying this statement

- L 472: but only in very rare cases (see Fig. 3A), this should be explicitly mentioned here

- L 480-81: where do the authors demonstrate that omnipresence of species is generally required? If they do, this should be referred to here

- L553: where do the authors show that positive interactions are more readily detectable than negative ones? (to the contrary, L359-60 reads: “There was no difference between the successful inference of positive interactions over negative interactions”)

- Fig. 1: may need re-arrangement (B comes before A)

Reviewer #2: Please find the review uploaded as a Word document.

**Have the authors made all data and (if applicable) computational code underlying the findings in their manuscript fully available?**

Reviewer #1: Yes

Reviewer #2: Yes

PLOS authors have the option to publish the peer review history of their article (what does this mean?). If published, this will include your full peer review and any attached files.

Reviewer #1: No

Reviewer #2: No
---

## [Decision Letter · Decision Letter 1]

23 May 2022

Dear Dr. Pinto,

Thank you very much for submitting your manuscript "Species abundance correlations carry limited information about microbial network interactions" for consideration at PLOS Computational Biology.

As with all papers reviewed by the journal, your manuscript was reviewed by members of the editorial board and by several independent reviewers. In light of the reviews (below this email), we would like to invite the resubmission of a significantly-revised version that takes into account the reviewers' comments.

The reviewer has asked for several clarifications and corrections, all of which I agree with. Further to those comments, I would recommend to shape the Introduction and Discussion to reflect extensive studies that have rather successfully used metabolic models in conjunction with co-occurrence analyses (binary as well as higher-order): e.g. Freilich et al. Nat Communications, 2011 and Machado et al. Nat Ecol Evol, 2021. The results from studies could be used to balance your discussion around the limits / benefits of correlational analyses.

We cannot make any decision about publication until we have seen the revised manuscript and your response to the reviewers' comments. Your revised manuscript is also likely to be sent to reviewers for further evaluation.

Sincerely,

Kiran Raosaheb Patil, Ph.D.

Deputy Editor

PLOS Computational Biology

The reviewer has asked for several clarifications and corrections, all of which I agree with. Further to those comments, I would recommend to shape the Introduction and Discussion to reflect extensive studies that have rather successfully used metabolic models in conjunction with co-occurrence analyses (binary as well as higher-order): e.g. Freilich et al. Nat Communications, 2011 and Machado et al. Nat Ecol Evol, 2021. The results from studies could be used to balance your discussion around the limits / benefits of correlational analyses.

Reviewer's Responses to Questions

**Comments to the Authors:**

Reviewer #1: The authors have improved, extended and clarified the manuscript. There are a few (minor) issues left, which if addressed could further improve the manuscript:

with regard to author response 2: “have not been systematically investigated against a ground truth” ... just for clarification: using mathematical models (gLV and others) as a ground truth has been done extensively in the literature (e.g. Weiss et al. (2016), also cited in the manuscript), albeit in variations that differ from what the authors did.

with regard to author response 3: “However, these techniques are also unable to recover directionality and type of interactions” ... SEMs/Bayesian networks do infer directed graphs (albeit with limitations), this sentence therefore seems misleading in the current form and requires either relevant references or proper explanation (similar to the response provided by the authors). Suggestion: “It is unclear, how well directed links predicted by these methods recover true ecological interaction types.”

with regard to author response 4: For clarification ... the given example constitutes not parasitism, but interference competition, where one species actively prevents another from obtaining resources used by both. The resulting interaction pattern would not be +/-, but rather -(weak)/-(strong), with one strong and one weak competitor. The original point stands: relationships in which one species simultaneously benefits from and harms another species are more likely to be stable if the exploitative benefit for species 1 outweighs the harm done to species 2. Observing unusual interactions with the opposite pattern, while low in number, should at least be briefly discussed, especially if they are used to support the central claim that interaction types are not detectable via. correlation methods. As a side note, perhaps the +/- class of interactions could be more adequately labeled “exploitative”, as they also include non-parasitic interactions such as predation.

with regard to author response 5: [regarding the adjusted abstract]: “limited information about the underlying web of interactions” sounds too strong, as many important characteristics of interaction networks (interaction signs, hub detection, various graph metrics) are either well recovered by correlations or have not been investigated by the authors. Suggestion: "limited information on specific interaction types” or “cannot distinguish detailed interaction types”. The line “competitive interactions may result in positive as well as negative correlations” is slightly misleading as is: requires quantification, as only very small fractions for competitive, commensal and amensal interactions had minority signs. Especially since this is already stated in the text (L. 530), it should for transparency also be in the abstract.

with regard to author response 7: While the addition to the methods section helps clarity, it is easy to overlook. It would therefore be useful also have a condensed reference to the parameterization challenges in the caveat section of the Discussion section.

**Have the authors made all data and (if applicable) computational code underlying the findings in their manuscript fully available?**

Reviewer #1: Yes

PLOS authors have the option to publish the peer review history of their article (what does this mean?). If published, this will include your full peer review and any attached files.

Reviewer #1: No
---

## [Decision Letter · Decision Letter 2]

5 Aug 2022

Dear Ms. Pinto,

Thank you very much for submitting your manuscript "Species abundance correlations carry limited information about microbial network interactions" for consideration at PLOS Computational Biology. As with all papers reviewed by the journal, your manuscript was reviewed by members of the editorial board and by several independent reviewers. The reviewers appreciated the attention to an important topic. Based on the reviews, we are likely to accept this manuscript for publication, providing that you modify the manuscript according to the review recommendations.

"...interactions are relatively rare among free-living bacteria and, if present, are often unidirectional. In contrast, Machado et al. (2021) suggested that"  In contrast is not fully incorrect since free-living and host-associated communities have been shown to have different patterns. I think removing "in contrast" should fix it.

Sincerely,

Kiran Raosaheb Patil, Ph.D.

Deputy Editor

PLOS Computational Biology

Kiran Patil

Deputy Editor

PLOS Computational Biology

[LINK]

"...interactions are relatively rare among free-living bacteria and, if present, are often unidirectional. In contrast, Machado et al. (2021) suggested that"  In contrast is not fully incorrect since free-living and host-associated communities have been shown to have different patterns. I think removing "in contrast" should fix it.

Reviewer's Responses to Questions

**Comments to the Authors:**

Reviewer #1: In this second revision, the authors have extensively answered to remaining criticisms. They have strengthened the manuscript by including further references and clarifications. I can now support its publication.

**Have the authors made all data and (if applicable) computational code underlying the findings in their manuscript fully available?**

Reviewer #1: None

PLOS authors have the option to publish the peer review history of their article (what does this mean?). If published, this will include your full peer review and any attached files.

Reviewer #1: No

Figure Files:

Data Requirements:

Reproducibility:

References:

---

## [Editor Report · Decision Letter 3]

15 Aug 2022

Dear Ms. Pinto,

We are pleased to inform you that your manuscript 'Species abundance correlations carry limited information about microbial network interactions' has been provisionally accepted for publication in PLOS Computational Biology.

Best regards,

Kiran Raosaheb Patil, Ph.D.

Deputy Editor

PLOS Computational Biology

---

## [Editor Report · Acceptance letter]

29 Aug 2022

PCOMPBIOL-D-22-00053R3 

Species abundance correlations carry limited information about microbial network interactions

Dear Dr Pinto,

I am pleased to inform you that your manuscript has been formally accepted for publication in PLOS Computational Biology. Your manuscript is now with our production department and you will be notified of the publication date in due course.

With kind regards,

Zsofia Freund
